# *VenoMS*—A Website for the Low Molecular Mass Compounds in Spider Venoms

**DOI:** 10.3390/metabo10080327

**Published:** 2020-08-11

**Authors:** Yvonne M. Forster, Silvan Reusser, Florian Forster, Stefan Bienz, Laurent Bigler

**Affiliations:** 1Department of Chemistry, University of Zurich, Winterthurerstrasse 190, 8057 Zurich, Switzerland; yvonne.forster@uzh.ch (Y.M.F.); stefan.bienz@chem.uzh.ch (S.B.); 2CAOS AG, Teufener Strasse 19, 9000 St. Gallen, Switzerland; silvan@caos.ch (S.R.); florian@caos.ch (F.F.)

**Keywords:** acylpolyamine, structure elucidation, HR-ESI-MS, MS/MS, fragment ion generation, *Agelenopsis aperta*

## Abstract

Spider venoms are highly complex mixtures. Numerous spider venom metabolites are uniquely found in spider venoms and are of interest concerning their potential use in pharmacology, agriculture, and cosmetics. A nontargeted ultra-high performance high-resolution electrospray tandem mass spectrometry (UHPLC-HR-ESI-MS/MS) approach offers a resource-saving way for the analysis of crude spider venom. However, the identification of known as well as the structure elucidation of unknown low molecular mass spider venom compounds based on their MS/MS spectra is challenging because (1) acylpolyamine toxins are exclusively found in spider and wasp venom, (2) reference MS/MS spectra are missing in established mass spectrometry databases, and (3) trivial names for the various toxin metabolites are used in an inconsistent way in literature. Therefore, we introduce the freely accessible MS website for low molecular mass spider venom metabolites, *venoMS*, containing structural information, MS/MS spectra, and links to related literature. Currently the database contains the structures of 409 acylpolyamine toxins, 36 free linear polyamines, and 81 additional spider venom metabolites. Implemented into this website is a fragment ion calculator (FRIOC) that allows us to predict fragment ions of linear polyamine derivatives. With three metabolites from the venom of the spider *Agelenopsis aperta*, it was demonstrated how the new website can support the structural elucidation of acylpolyamines using their MS/MS spectra.

## 1. Introduction

More than 98% of the components of untargeted metabolomics’ experiments remain uncharacterized because the reference spectra are not available in established mass spectrometry databases (*National Institute of Standards and Technology* (*NIST)*, *Metlin*, *MassBank*, *Global Natural Product Social Molecular Networking (GNPS)*, and others) [1]. This makes the investigation of natural products challenging. The study of spider venoms, containing large molecules such as enzymes and peptides and low molecular mass compounds, namely free polyamines and acylpolyamines as well as other small compounds such as biogenic amides, amino acids, or nucleosides, is no exception. In particular, the identification of acylpolyamine toxins, previously found exclusively in the venoms of spiders and wasps [2], is difficult because hardly any representatives of this compound class can be found in established databases.

Over the past 20 years, our group has devoted itself to the structure elucidation of low molecular mass compounds in spider venoms and, in particular, the study of acylpolyamines by HPLC-MS and MS/MS. Acylpolyamine are a class of neuroactive compounds, generally active on Ca^2+^ channels, K^+^ channels, and ionotropic receptors. They are of particular interest for their potential use in drug discovery, especially in relation to brain diseases [3,4,5,6]. Such polyamine toxins (Figure 1) consist usually of an aromatic acyl head, a polyamine backbone of varying length, and an alkaline tail. Optionally, an amino acid linker, such as asparagine, can be part of the molecule, and the amines can be substituted with hydroxy or methyl groups.

Historically, the nomenclature of acylpolyamines mostly refers to the originating spider species and the molecular mass of the toxins. As a result, several toxins obtained five or even more trivial names. Isomeric structures demanded the extension of the trivial names with additional letters, which was done randomly. All this makes literature search extremely laborious and impedes the cross-linking of results from different research groups. Alternatively, a generic nomenclature was proposed for such compounds [7]. It uses abbreviations for the head, tail, and amino acid linker groups, and the polyamine backbones are denoted by numbers representing the numbers of methylene units in between two N-atoms. A substituent at an N-atom is indicated in brackets. For example, the generic name of the structure shown in Figure 1 is 2,4-(OH)_2_-PhAcAsn533Arg while the acylpolyamine shown in Figure 2 is called 4-OH-Bz3(OH)334. Even though the generic names are slightly longer, they are unique and provide structural information, which is a significant improvement compared to the use of trivial names.

Our group started with the structure elucidation of acylpolyamines in spider venoms by HPLC-MS and MS/MS [7,8], improved the analytical methods in order to obtain better selectivities and sensitivities [9,10], synthesized acylpolyamines as reference material [11,12,13], and studied their MS/MS fragmentation behavior [14]. The key revelation of these studies was that polyamine derivatives undergo mainly intramolecular nucleophilic substitution reactions at protonated amino groups under MS/MS conditions. This resulted in a handful of fragmentation rules specific to acylpolyamines and a nomenclature describing the fragment ions of polyamines [15]. These rules are summarized on the homepage under (>“*Analytical tools*” >“*Fragmentation rules for acylpolyamines*”). As an example, the structure of the synthetic acylpolyamine 4-OH-Bz3(OH)334 and its MS/MS spectrum are shown in Figure 2, with the fragmentation positions and the respective ion responses labeled accordingly. 

Rule-based structure elucidation can be improved by comparing the spectrum of an unknown analyte with the MS/MS spectra of structurally related compounds. Literature research on previously found metabolites in related spider species can also provide clues. However, a database that compiles low molecular mass spider toxins and their properties is missing up to now [16]. For this reason, *venoMS* (www.venoms.ch) was developed, a well-structured website containing a compilation of the structures found in spider venom toxins. The website focuses on spider venom metabolites with low molecular masses (<1000 Da), is freely accessible online, and can be considered as complementary to *ArachnoServer* [17], the database providing detailed information about origin, structure, and biological activity of protein toxins in spider venom. *VenoMS* covers all low molecular mass spider venom components known to date and provides links to literature relating to structure elucidation, synthesis, and biological activity. The acylpolyamines are listed under their generic names, which will simplify the comparison of data sets associated to the structure of polyamine toxins. In addition, *venoMS* contains standardized compound information including LC-MS, MS/MS, and hydrogen/deuterium-exchange (HDX) data. A new fragment ion calculator (FRIOC) for polyamine toxins was implemented in order to support the identification of known as well as the structure elucidation of yet unknown acylpolyamines on the basis of their MS/MS spectra. With fragmentation rules of acylpolyamines at hand, the structures of polyamine toxins can now be derived in most cases from MS/MS spectra without the need for tedious syntheses of reference materials.

## 2. Results

### 2.1. The Website

The *venoMS* is, to the best of our knowledge, the first freely accessible website containing the description of low molecular mass compounds (<1000 Da) found in spider venoms. It can be accessed at www.venoms.ch. Information on the structure and the biological function of the metabolites has been compiled together with the corresponding MS/MS spectra, annotation of the fragment ions, and links to related literature data. The *venoMS* was developed as a static html documentation page generated with *hugo* [18], a popular web site generator used for building websites based on a content management system (cms). The programming language of *hugo* is *go*, which was originally created by *Google* in 2012 [19].

#### 2.1.1. Data Curation

The setup of the website was started with an extensive literature search including reviews on low molecular mass compounds found in spider venom [2,3,4,5,6,20,21,22,23,24,25,26,27,28,29,30]. The *SciFinder* and *Google Scholar* search platforms were screened for references related to “spider venom” and the results were refined with the terms “acylpolyamine”, “acetylpolyamine”, “low molecular compound”, or “small compound”. All hits were added to the database after examination of the respective original literature on *SciFinder*. For the class of acylpolyamine derivatives, all types of publications including structure elucidation, synthesis, biological activity, and review articles were included.

Since venoms, in general, contain small molecules that also play an important role in a multitude of biological processes, the literature search for free polyamines and other small compounds was limited to publications related to spider venoms. Overall, 263 publications relating to more than 300 low molecular mass compounds were considered (Figure 3).

#### 2.1.2. Levels of Reliability of Data and Structure Identification

The techniques applied for the structure elucidation of compounds found in spider venoms vary substantially. In the best case, the metabolite structures were unambiguously proven by the synthesis of reference material. Other structures were determined by interpretation of NMR and/or MS/MS spectra. However, some of the published toxin structures were only suggested due to their mass and the assumption that acylpolyamines are composed of already known units. For this reason, the entries included in *venoMS* were classified into the structure reliability levels S-1–S-5, taking into account the broad range of the analytical methods used for the structural analysis and their reliability. **S-1** Structure confirmed with the synthesis of reference material and analyzed with state-of-the-art analytical methods.**S-2** Structure elucidated from NMR data obtained from collected venom.**S-3** Structure elucidated from the interpretation of MS/MS data.**S-4** Structure deduced from other analytical techniques such as full scan MS data (FS-MS), fast-atom bombardment (FAB), UV absorption, and/or amino acid analysis.**S-5** Unknown, nonconfirmed structure. The compound VdTX1, for example, has only been described as an HPLC peak with retention time (*R*_t_) 22.2 min, with a mass of mass-to-charge ratio (*m/z*) 729.406, and with a reversible neuronal blockade activity [31].

In order to improve the quality assessment of structure elucidation with methods based on HR-MS/MS, confidence scales have been introduced such as those proposed by *Schymanski* et al. [32] or *Oberacher* et al. [33]. However, due to the scarcity of synthetic reference material that could have been measured with our standardized analytical method (SAM2020, see below) and to the lack of MS/MS spectra of polyamine derivatives in commercially or free accessible databases, none of the existing systems seemed suitable for our purposes. It was, therefore, decided to create a confidence scale with the following categories C-1 to C-4, classifying the structures of the polyamine derivative according to the available analytical data:**C-1** MS/MS data acquired with the standardized analytical method (SAM2020) published on www.venoms.ch (“Analytical tools” > “Analytical method”) available.**C-2** Link to MS/MS data acquired with a different analytical method available.**C-3** No recorded MS/MS spectrum available. The mass of potential fragment ions generated by the fragment ion calculator (FRIOC) for polyamine derivatives (see below) are displayed.**C-4** No recorded MS/MS spectrum nor generated fragment ions available, e.g., the neuroactive glyconucleoside disulfate HF-7 [34].

#### 2.1.3. Fragment Ion Calculator (FRIOC) for Polyamine Toxins 

The *venoMS* website was supplemented with *FRIOC*, an MS/MS fragment ion calculator for acylpolyamines and other polyamine derivatives. *FRIOC* is based on fragmentation rules for acylpolyamines [15], supporting a straightforward structure elucidation. It is a server-based calculation engine based on the popular programming language *go* [19]. The source code is hosted on *github* [35], where it automatically creates releases and deploys to a *Google* Cloud-run instance for the usage through third parties.

The *FRIOC* calculator tool can be easily accessed through the *venoMS* website header. Similar to the generic nomenclature of acylpolyamines, it is assumed that a polyamine toxin consists of a head, of one to 10 polyamine backbone units, and of a tail, each of which is linked to the previous unit via a nitrogen atom (Figure 1). An acylpolyamine structure can be easily assembled by selecting individual units from drop-down menus (Figure 4).

FRIOC then generates the generic name of the compound and calculates its chemical formula, molecular mass, and number of exchangeable protons (HDX), as well as the mass of the ionized precursor molecules and related rule-based MS/MS fragment ions (a-, b-, c-, t_a_-, t_z_-, y-, and z-type ions). Data generated by FRIOC can also be found in tabular form in each (acyl)polyamine entry registered in *venoMS* and can easily be compared to acquired MS/MS data (see Figure 5).

#### 2.1.4. Standardized Analytical Method (SAM2020)

Retention times (*R*_t_s) and ESI-MS/MS spectra of an analyte depend strongly on the used instrumentation and the selected settings. Therefore, validation of a structure elucidation can only be done by comparison of data acquired under the same analytical conditions. For this reason, all spider venom samples available to us were analyzed using a single analytical method, i.e., the same chromatographic conditions, MS parameters, and stepped normalized collision energy (30, 35, 40), resulting in the standardized *R*_t_s and ESI-MS/MS spectra contained in the *venoMS* website.

Since spider venoms were only available in limited amounts, the focus of the MS/MS acquisition method has been placed on a high duty cycle and an optimal collision energy for polyamine derivatives instead of data collected at different collision energies as recently recommended by the metabolomics community [36]. This results in a highest sensitivity and a better distribution of fragment ions over the entire mass range and simplifies the visual comparison between measured spectra and those obtained from the FRIOC fragmentation tool (see above).

The analytical method was supplemented with the acquisition of UHPLC-HR-MS data in the full scan mode performed with a mobile phase containing deuterated solvents (MeCN/D_2_O + 0.05% mono-deuterated trifluoroacetic acid (d_1_-TFA)). Since the polar compounds of interest elute at high D_2_O content being greater than 75%, distinct mass shifts of the ionized molecules were obtained, allowing unambiguous determination of the number of exchangeable hydrogen atoms. This strongly increases the reliability of the structure elucidation, in particular with regards to the differentiation of constitutional isomers [9]. The use of HDX in the determination of the number of exchangeable protons in (acyl)polyamine structures has been implemented in the FRIOC calculator.

All details of the analytical method can be found in Section 4.3 and on the *venoMS* website under “*Analytical tools*” > “*Analytical method*”.

## 3. Discussion

### 3.1. Functionalities of the Website

The website was created to support the annotation of known and the structural elucidation of unknown compounds in spider venoms. It contains a list of all low molecular mass metabolites found so far in spider species. The website can be searched (by keywords) for metabolites with a specific mass or molecular formula, and it allows sorting the entries by individual spider species and families. This way it can, for instance, be recognized at a glance that in spiders of the family of *Araneidae* mainly acylpolyamines possessing an amino acid linker have been found, whereas in spiders of the family of *Agelenidae* only acylpolyamines without such an amino acid linker have been detected so far. Finally, almost no polyamine toxins have been found in spiders of the *Lycosidae* family.

Each database entry includes in addition to the spider species where the structure was found, a general description including name, chemical formula, molecular mass, CAS number, number of exchangeable hydrogen atoms (HDX), the corresponding reference literature, masses of precursor ions, masses of measured MS/MS fragment ions, masses of fragment ions generated by *FRIOC*, and *R*_t_ if the standardized method (SAM2020) has been used, and, finally, the level of identification reliability and category of comparability. 

Getting the structure of a defined compound with known CAS number by using *SciFinder*^®^ is not an issue. It is difficult, however, to come to reliable structure of (acyl)polyamine candidates on the basis of limited analytical data and to confirm their presence (including literature data), or to determine the structure of an unknown compound. Exactly to solve this problem for polyamine derivatives is the purpose and the innovation of *venoMS*. This will be illustrated with the following three examples.

### 3.2. Identification of Individual Compounds—General Workflow

The use of the *venoMS* website for the structural elucidation of metabolites in spider venom relies mainly on a chemical formula. The calculation of unequivocal chemical formulae is, therefore, a key issue and only possible with the acquisition of accurate masses and by considering, e.g., adducts, isotopic distributions, and/or fragment information. For this reason, we filtered the chemical formulae according to the Seven Golden Rules, which is strongly recommended to be done also by the users of *venoMS* [37]. A simple way is to process the mass spectra with embedded software from instrument manufacturers, such as, e.g., FreeStyle^TM^ (ThermoFisher Scientific, Waltham, MA, USA), iFit^TM^ (Waters, Wilmslow, UK), or SmartFormula 3D^TM^ (Bruker Daltonics, Bremen, Germany).

The flow chart in Figure 6 summarizes how the structure of a venom constituent can be determined from an LC-MS experiment considering *R*_t_ and mass-to-charge ratio (*m/z*). The procedure depends on whether the analyte substance is a known compound, of which a reference spectrum is stored in the database (queries 1–3, abbrev. Q1–Q3, see example 3.2.1. below), the candidate is a known polyamine toxin for which no reference spectrum is available (Q1–Q7, example 3.2.2.), or the acylpolyamine is unknown and its structure needs to be elucidated, which can be done assisted by FRIOC that allows the manual assembly of possible candidates (Q1, Q8–Q10, example 3.2.3). In this case, the consistency of the proposed structure can be scrutinized in the same manner as that of a known acylpolyamine archived without its MS/MS spectrum (Q5–Q7).

In the ideal case, a spider venom was investigated by means of the standardized analytical method (SAM2020) reported on www.venoms.ch (>“*Analytical tools*” > “*Analytical method*”). In this case, reliable comparative information regarding *R*_t_, relative intensities of fragment ion signals, and numbers of exchangeable protons (HDX) can be found on the website. If a different method and/or a different instrumentation was used, it is important to keep in mind that the chromatographic behavior (*R*_t_ values) of the candidates and the relative intensities of the signals in their MS/MS spectra may significantly differ from those reported in *venoMS*. 

The numbered queries of the workflow are explained in more detail as following:


**Q1. Has a compound with a specific chemical formula previously been found in spider venoms?**


The search query for the chemical formula returns a corresponding list of all known substances. Alternatively, the website can be explored for the mass of singly or doubly charged precursor ions (rounded down to the nearest whole number). The candidates must then be evaluated one by one (Q2). If no hit is found, the investigated mass might correspond to adduct ions, to in-source collision induced dissociation (IS-CID), or might be related to a new compound (Q-8). 


**Q2. Is the observed retention time (*R*_t_) reasonable?**


The *venoMS* includes absolute *R*_t_ values obtained from polyamine spider toxins analyzed with the standardized method (see on www.venoms.ch under “*Analytical tools*” > “*Analytical method*”). Using the same method, the measured *R*_t_ of an analyte should not deviate by more than 2% from the *R*_t_ of a potential candidate found in *venoMS*. Q2 shall be skipped if the *R*_t_ value is not suited as comparative data (e.g., because a different method was used). It is also noteworthy that the composition of a venom might have a substantial and unpredictable influence on the *R*_t_ values of its individual components due to matrix effects. Nevertheless, we have found that *R*_t_ values are important and helpful descriptors, especially when they are considered relative to each other. For example, the *R*_t_ values of acylpolyamine derivatives increase in relation to an increasing number of N-hydroxyl groups in the polyamine chain. In addition, the range of *R*_t_ values gives first hints for the head group structure. Finally, an alleged free polyamine candidate with a *R*_t_ > 2 min corresponds most likely to an acylpolyamine derivative precursor undergoing ion-source CID.


**Q3. Is an MS/MS reference spectrum available and does it match?**


If a reference MS/MS spectrum of a proposed analyte is available, it can easily be compared with the experimental data. If the spectra match, the substance can be considered as identified. To get even more confidence, it is possible to check whether the experimentally determined number of exchangeable H-atoms corresponds to the expectations (Q7). If no reference spectra are available for the proposed analyte, the further steps depend on whether the compound is a polyamine derivative or another small compound (Q4).


**Q4. Is a polyamine derivative suspected or another small compound?**


The structure of polyamine toxins can be further assessed in *venoMS* (Q5). In the case of small compounds, it is recommendable to screen established databases (e.g., *Metlin*, *MassBank*, *Human Metabolome Database (HMDB)*, and *ChemSpider*) for possible candidates. 


**Q5. Are the fragment ions generated with *FRIOC* matching?**


The presence of MS/MS signals calculated by *FRIOC* for fragment ions is a good indication for the suggested substance; in particular, if the signals relate to a- and t_a_-type ions. At this point, it is not necessary that all rule-based fragment ions are found in the MS/MS spectrum of a candidate to include it into a ‘preliminary’ hit list. Additional confidence can be gained by answering the questions Q6 and Q7.


**Q6. Does the entire MS/MS fingerprint match the signals in the spectrum of a structurally related compound?**


For the structural elucidation of acylpolyamines by MS/MS, the relative ion intensities have also to be considered. However, they not only strongly depend on the collision energy chosen but also on the type of instrument used and the collision gas (e.g., Ar, N_2_, or He). Furthermore, signal intensities cannot be predicted by *FRIOC* so far. Since ion patterns are mainly influenced by the polyamine backbone, it makes sense to compare the MS/MS spectrum of a suggested structure with the spectrum of a compound with an identical polyamine backbone. If such a corresponding MS/MS spectrum is not available, it might finally be useful to compare the experimental data with the spectrum of a closely related structure with a different chemical formula. 


**Q7. Does the number of observed exchangeable protons agree with the theoretical number?**


The determination of the number of exchangeable protons can give further confirmation of the proposed structure. This number is particularly helpful for recognizing and locating substituted and quaternary amines. 

If one or several of the above criteria Q4 to Q7 are not fulfilled, a suggested structure can be excluded, and a next candidate should be reviewed (back to Q2). If none of the listed compounds fits, the investigated spectra might be related to a substance that has not yet been described in spider venom. Then continue with Q8.


**Q8. Is the alleged compound a polyamine toxin?**


The chemical formula, the number of N-atoms, and the calculated ring and double bond equivalents are good indicators for deciding whether it is worth to focus on polyamine structures and to proceed with *venoMS* and to go to the next step (Q9) or to focus on other small molecules (see Q4).


**Q9. Identify the head moiety from MS/MS data.**


To establish the structure of a new polyamine toxin, it is absolutely necessary to identify in a first step the head unit of the compound. Fortunately, most head and some tail groups lead to characteristic fragment ions by which they can be unambiguously identified. A list of these diagnostic ions is available in the *venoMS* website under “Analytical tools” > “Characteristic fragment ions”. If the head group cannot be identified, it may still be possible to elucidate the structure using the MS/MS spectrum. However, *FRIOC* cannot be used because the structure must be assembled from head to the tail with the confirmed structural units implemented into the tool. It should further be noted that *FRIOC* is limited to already known units. New units can be implemented by contacting us at venoms@chem.uzh.ch.


**Q10. Use *FRIOC* to generate the fragment ion pattern of a presumed acylpolyamine.**


Once the head portion is identified, the new acylpolyamine can be assembled step by step with the easy-to-use *FRIOC* tool. When the assembled acylpolyamine matches the mass of the precursor, the new structure can be validated again via Q5–Q7. 

### 3.3. Examples of Compound Identification

The three following examples were chosen to demonstrate how *venoMS* and *FRIOC* can be used for the annotation of known and the structure elucidation of unknown polyamine toxins according to the general workflow presented in Figure 6. All three compounds were found in the venom of the spider *Agelenopsis aperta*, and the experimental data were recorded with the standardized analytical method reported in *venoMS*.

#### 3.3.1. Identification of a Known Acylpolyamine (Confidence Level S-1) with Available MS/MS Reference Spectrum (Comparison Level C-1)

The descriptors of the peak of interest were a mass of *m/z* 417.33334 and the corresponding chemical formula (C_23_H_41_N_6_O^+^, −0.72 ppm) and a retention time of *R*_t_ 8.89 min. **Q1.** Searching the database with the chemical formula C_23_H_40_N_6_O yielded four hits, IndAc3334, IndAc3343, IndAc3433, and IndAc4333. All four toxins have been previously described in the venom of *A. aperta*.**Q2.** The *R*_t_ values for IndAc3334 (7.59 min), IndAc3343 (7.64 min), and IndAc3433 (7.82 min) deviated significantly from the observed 8.89 min, and the respective compounds were, therefore, excluded. Only the *R*_t_ value for IndAc4333 (9.00 min) was within the accepted range of 8.89 ± 0.18 min and this compound was, therefore, taken to the next step.**Q3.** The MS/MS spectrum of the analyte perfectly matched the spectrum of synthetic IndAc4333 stored in *venoMS* (Figure 7). The alleged compound was thus unambiguously identified as IndAc4333.

#### 3.3.2. Identification of a Known Acylpolyamine (Confidence Level S-1) without a Reference MS/MS Spectrum (Comparison Level C-3)

In the second example, an ion of *m/z* 505.38687 was found at *R*_t_ 8.10 min, leading to a presumed chemical formula of C_27_H_49_N_6_O_3_^+^ (1.58 ppm). Interestingly, the base peak signal observed in the full scan spectrum (FS-MS) corresponded to the doubly charged ion (*m/z* 253.19699, C_27_H_50_N_6_O_3_^2+^, [M + H]^2+^, 1.26 ppm). In addition, a TFA adduct was detected at *m/z* 619.37962 (C_29_H_50_N_6_O_5_F_3_^+^, [(M + H) + CF_3_CO_2_]^+^, 1.11 ppm).**Q1.** The database search using the chemical formula of the nonprotonated structure (C_27_H_48_N_6_O_3_) delivered no hit. The alternative query, searching for a charged precursor ion P^+^ = *m/*z 505 revealed a single hit: 4-OH-IndAc3(OH)335(NMe_3_)^+^.**Q2.** This singly charged acylpolyamine had previously been described as a component of the venom of *A. aperta*, and its structure was confirmed by synthesis [38]. However, this compound has never been analyzed before with the standardized method. Therefore, *venoMS* offered no *R*_t_ value for this compound to be compared.**Q3.** The *venoMS* did not contain any reference MS/MS spectrum for the proposed structure at the time of its analysis.**Q4.** The proposed structure 4-OH-IndAc3(OH)335(NMe_3_)^+^ corresponds to an acylpolyamine and can be further scrutinized.**Q5.** For polyamine toxin included in the database that have no reference MS/MS spectrum, *venoMS* provides a table with potential fragment ions related to the proposed structural moieties that were calculated by *FRIOC*. Consequently, the recorded MS/MS spectrum (Figure 8, top) was compared to the calculated fragment ions. The ions *M*^+^ (*m/z* 505.38680, 1.44 ppm), **a’** (*m/z* 146.06041, 2.53 ppm), **a_1_** (*m/z* 231.11341, 2.64 ppm), **a_2_** (*m/z* 304.16639, 2.70 ppm), **a_3_** (*m/z* 361.22422, 2.21 ppm), and **a_4_** (*m/z* 446.31402, 3.25 ppm) matched well with the ion masses provided by FRIOC. Although none of the other calculated fragment ions were observed (dashed lines in Figure 8, bottom), the presence of the a-type fragment ions is a strong argument for the proposed polyamine toxin.**Q6.** Screening the database for acylpolyamines with an identical polyamine backbone (3(OH)335(NMe_3_)) revealed that the only structurally related entry was IndAc3(OH)335(NMe_3_)^+^. This compound was found in the venom of *A. aperta* and its structure was verified at a level S-1, meaning by synthesis. [38]. However, at the time of the analysis, the database did not contain any MS/MS spectrum for IndAc3(OH)335(NMe_3_)^+^ that could have been used for comparison (comparability level C-3).**Q7.** Finally, on-column hydrogen/deuterium (H/D) exchange LC-MS revealed the presence of six exchangeable H-atoms in the examined compound (Figure 9, top), eliminating the possibility of the presence of a dimethylammonium tail and an additional methylene group within the polyamine backbone. It can, therefore, be concluded that the structure of the investigated venom component corresponds to 4-OH-IndAc3(OH)335(NMe_3_)^+^.

#### 3.3.3. Structure Elucidation of an Unknown Acylpolyamine (Confidence Level S-3, Comparison Level C-3)

The signal of the compound of interest was recorded with a mass of *m/z* 491.37039 at *R*_t_ 8.32 min. The accurate mass and the isotopic distribution led to the chemical formula C_26_H_47_N_6_O_3_^+^ (−0.06 ppm).

**Q1.** The molecular formula (C_26_H_46_N_6_O_3_) did not match with any database entry. The search for the precursor mass P^+^ at *m/z* 491 resulted in the single hit, 4-OH-IndAc3(OH)334Gu. However, the accurate mass of this acylpolyamine deviated roughly by 50 ppm from the measured value of the protonated molecule, which excludes its suggestion as a sensible candidate.

**Q8.** The ratio of N-, C-, and H-atoms in the molecular formula C_26_H_46_N_6_O_3_ suggested that the compound could belong to the class of acylpolyamine derivatives.

**Q9.** The MS/MS (Figure 9, bottom) was then screened for characteristic fragment ions revealing the head moiety of the potential acylpolyamine. For support, a list of the potential **a’**, **a_0_**, and **a_1_** type fragment ions of head moieties is given in the *venoMS* website section “ *Characteristic fragment ions”*. In this case, the fragment ions *m/z* 231.11307 (**a_1_**, 1.17 ppm) and *m/z* 146.06014 (**a’**, 0.68 ppm) were identified, suggesting the presence of the *4-OH-IndAc* aromatic head linked to a propylamine unit.

**Q10.** The identified head moiety was used as the starting point to assemble, step by step, new acylpolyamine candidates. By focusing on a-type fragment ions (*m/z* 304.16583 (**a_2_**, 0.85 ppm), *m/z* 361.22346 (**a_3_**, 0.11 ppm), and *m/z* 446.31320 (**a_4_**, 1.41 ppm), **Q5**), the acylpolyamine was identified as an 4-OH-IndAc3(OH)335* derivative with an unknown tail moiety. The accurate mass difference between the precursor ion and the **a_4_** fragment ion revealed the neutral loss of C_2_H_7_N (45.05836 Da, 11.34 ppm). This chemical formula did not correspond to any formerly known tail unit and needed further investigation.

**Q6.** A database search for compounds with a structurally related polyamine backbone revealed that the signal intensities of the fragment ions of 4-OH-IndAc3(OH)335(NMe_3_)^+^ were almost the same as those of the analyte ions (Figure 9). This means that the two structures most possibly differ only by their tail part.

**Q7.** The neutral loss of C_2_H_7_N allowed only two possible structures for the tail moiety of the analyte: *N(Me_2_) or *NH(CH_2_CH_3_). The two possible structures can be distinguished by the total number of exchangeable H-atoms, 7 for the *N(Me_2_) derivative and 8 for *NH(CH_2_CH_3_), respectively. The on-column H/D exchange experiment resulted in a mass shift of seven units, of which six exchangeable protons could be derived from the alleged 4-OH-IndAc3(OH)335 partial structure, two to the head and four to the polyamine backbone. The deuteration resulting from the ionization is the reason for the last mass shift. Since the tail moiety does not contain a further exchangeable H-atom, the tail group *NH(CH_2_CH_3_) was excluded and *N(Me_2_) remained as the only possible structure of the tail. The final structure of the unknown acylpolyamine was, therefore, concluded to be 4-OH-IndAc3(OH)335(NMe_2_).

## 4. Materials and Methods

### 4.1. VenoMS Website

The programming language is *go*. The website design was adapted from the *hugo* theme *docdock* (https://github.com/vjeantet/hugo-theme-docdock). The website is hosted on *github* pages. It is an open source project with Apache License 2.0, which can be accessed at https://github.com/ms-uzh/venoms.

### 4.2. Fragment Ion Calculator (FRIOC) for Polyamine Derivatives 

The server-based calculation engine is written in the programming language *go*. The source code is hosted on *github* and can be accessed at https://github.com/ms-uzh/calc (Apache License 2.0). The calculations are executed in a Google Cloud-run project.

### 4.3. UHPLC-HR-ESI-MS and MS/MS Method

The UHPLC-ESI-MS and MS/MS experiments were performed on a Dionex UltiMate 3000 HPLC instrument (Thermo Scientific, Germering, Germany) equipped with an autosampler, a pump, and a diode-array detector (DAD) of the same producer series. The UHPLC system was connected to a QExactive Orbitrap FT mass spectrometer (Thermo Scientific, Waltham, MA, USA) equipped with the heated ESI source.

The samples—lyophilized spider venom dissolved in H_2_O/MeCN 3:1 + 0.05% TFA to a concentration of 1 μg/μL—were chromatographed at a flow rate of 0.30 mL/min on a RP-C_18+_ column (Cortecs UPLC C_18+_ 1.6 μm, 2.1 mm × 150 mm, Waters, Milford, MA, USA) with solvents A and B consisting of H_2_O + 0.05% TFA and MeCN/H_2_O 8:2 [*v*/*v*] + 0.05% TFA, respectively. The column chamber temperature was set to 25 °C. The samples were injected at a volume of 1 μL. The strong wash was MeOH and weak wash H_2_O/MeCN 95:5 [*v*/*v*]. The gradient increased from 3.8 to 25% of eluent B over 22 min. Finally, the column was regenerated at 100% B for 4 min and equilibrated at 3.8% B over 6 min.

The MS conditions were: Sheath gas flow rate (N_2_, 48), aux gas flow rate (N_2_, 11), aux gas heater temperature (320 °C), sweep gas flow rate (N_2_, 2), spray voltage (3.5 kV), S-lens RF level (55), and capillary temperature (256 °C). Full scan MS acquisitions were performed in positive ionization mode at 70,000 resolutions and ions were detected in the mass range between *m*/*z* 100 to 1500. The ion gauge control (AGC) target setting for full scan MS experiment was set to 10⁶ with a maximum of 30 injection times. The mass spectrometer was calibrated for mass accuracy once a day according to the manufacturer’s instructions. The relative mass error being typically lower than 2 ppm.

The MS/MS experiments were acquired in alternated Full MS/data dependent data-depended MS/MS mode (Top *N* = 5) with a resolving power of 17,500. The acquisition was performed with an isolation window set to 0.8 *m*/*z*. The minimum AGC target setting was set to 10^3^. A stepped normalized collision energy (NCE) was used (30, 35, 40).

For the H/D exchange experiment, the mobile phase of the above described UHPLC-ESI-MS method was changed and solvents A and B, consisting of D_2_O + 0.05% d_1_-TFA and MeCN/D_2_O 8:2 [*v*/*v*] + 0.05% d_1_-TFA, respectively, were used. The column chamber temperature was set to 40 °C. All the other parameters remained the same as under normal conditions.

## 5. Conclusions

The platform www.venoms.ch comprising the *venoMS* website and the polyamine fragment ion calculator *FRIOC* is the first freely accessible website for MS-based structure elucidation and identification of low molecular mass compounds present in spider venom. The website comprises a compilation of very heterogenous data from literature and of standardized data from our own investigations that allows the efficient identification of known and already characterized low molecular mass compounds contained in spider venom. It encloses MS/MS spectra of all spider venom metabolites (<1000 Da) known to date, together with links to the corresponding literature. The implemented fragment ion calculator FRIOC is a tool to construct hypothetical polyamine toxins from known building blocks and to calculate accurate masses and the ion distributions to be expected for such compounds. It straightforwardly allows reviewing structural proposals by comparison of predicted ion distributions with experimental data. The *venoMS* and FRIOC do not contain functionalities that do automated MS analysis and structural elucidations but they offer tools and procedures to efficiently identify and structurally elucidate low molecular mass compounds of a specific natural product class.

The website does not allow open access data upload. However, since a clear focus on structure identification and elucidation of polyamine derivatives based on MS/MS is recognizable and less than 0.5% of the known spider species have been screened for their polyamine derivatives so far, progress in the field of spider venom characterization by MS/MS is to be expected and the website needs to be serviced and maintained. This will be done by us. We will supplement the database in the future with new structures, related MS/MS spectra, and structural units obtained from future studies. Additional information that is not directly accessible within the website, e.g., biological activities or syntheses, will be made accessible by links to related literature also in future. It is planned to expand the content of *venoMS* with this type of information in a next step. 

Since *venoMS* was created in a rather generic way, the same basic website construction could be used for the creation of other specialized databases for the analysis of other products that show rule-based fragmentation behaviors.

## Figures and Tables

**Figure 1 metabolites-10-00327-f001:**
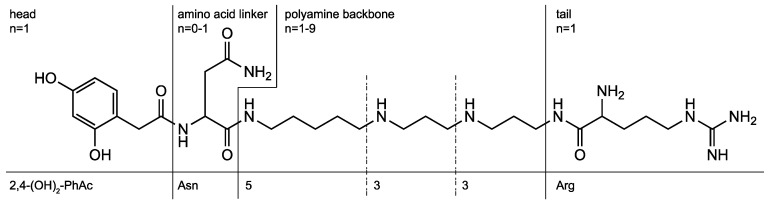
Polyamine toxins consist of an acyl head, a polyamine backbone, an alkaline tail, and an optional amino acid linker. The structure shown is known by the trivial names AR 636, Arg 636, argiopine, argiotoxin 636, ArgTX 636, ATX, and AVTX 636. The generic name is 2,4-(OH)_2_-PhAcAsn533Arg.

**Figure 2 metabolites-10-00327-f002:**
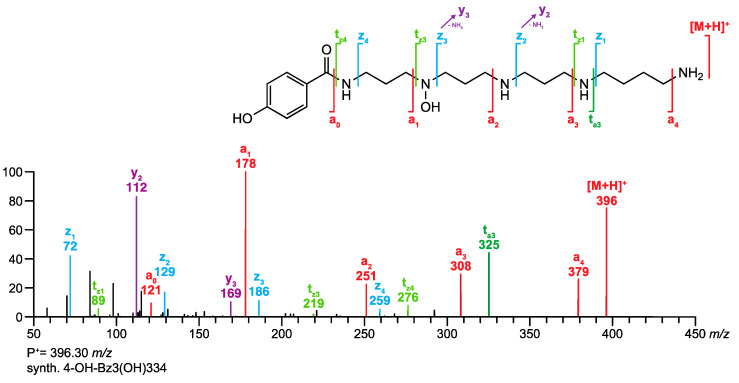
MS/MS spectrum of synthetic 4-OH-Bz3(OH)334 and annotation of the fragment ions according to the fragmentation rules of acylpolyamines introduced by *Tzouros* et al. [11].

**Figure 3 metabolites-10-00327-f003:**
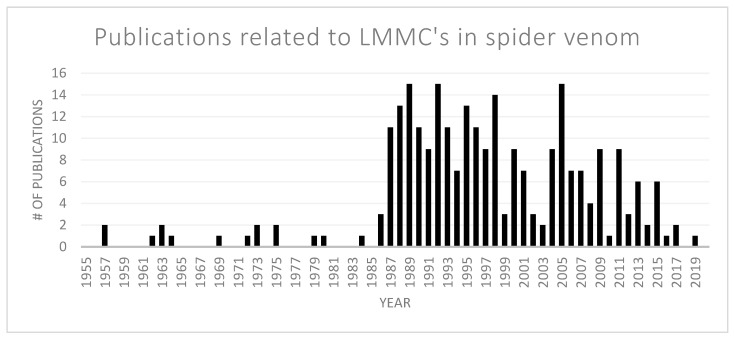
Number of publications related to low molecular mass compounds (LMMCs) described in spider venom from 1955 until September 2019. The breakthrough in this research field came at the end of the 1980s with the structural elucidation of the first acylpolyamines.

**Figure 4 metabolites-10-00327-f004:**
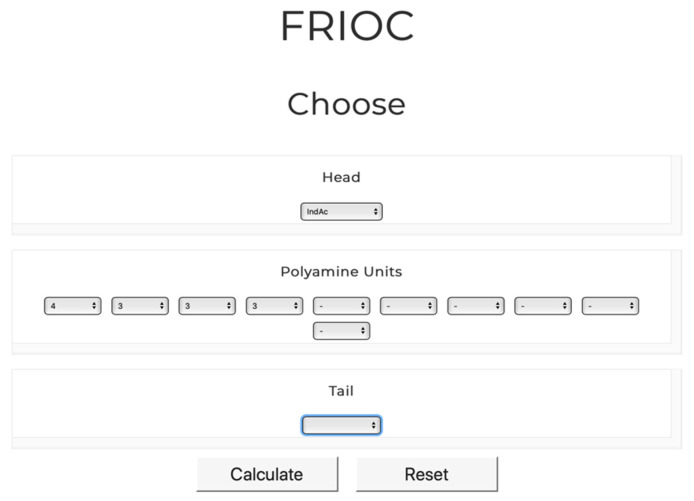
FRIOC interface used to assemble a new acylpolyamine structure and generate the corresponding MS/MS spectrum. The output obtained for IndAc4333 is shown in Figure 5.

**Figure 5 metabolites-10-00327-f005:**
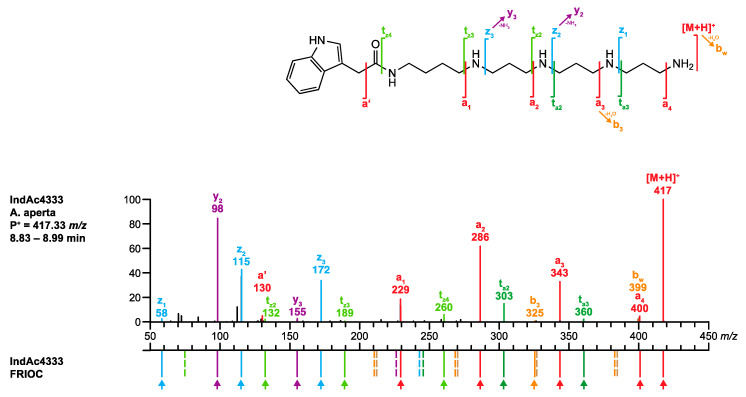
Comparison of the MS/MS spectrum of IndAc4333 found in the venom of *A. aperta* with the spectrum calculated by the FRIOC tool. Spectra as well as measured and calculated values are shown in Appendix A.

**Figure 6 metabolites-10-00327-f006:**
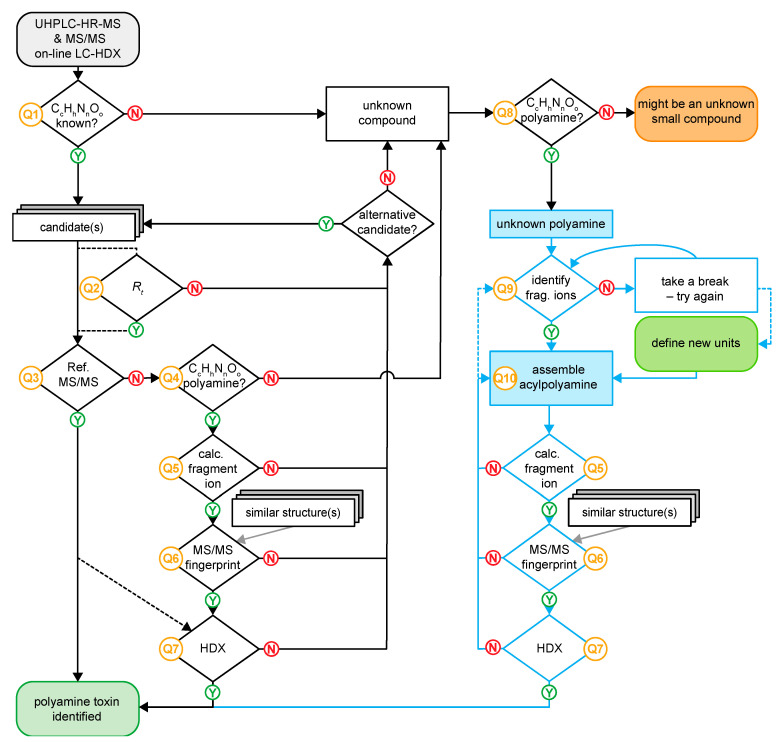
Flow chart of *venoMS* allowing the identification of a known low molecular mass compound from spider venom or the structure elucidation of a new polyamine toxin.

**Figure 7 metabolites-10-00327-f007:**
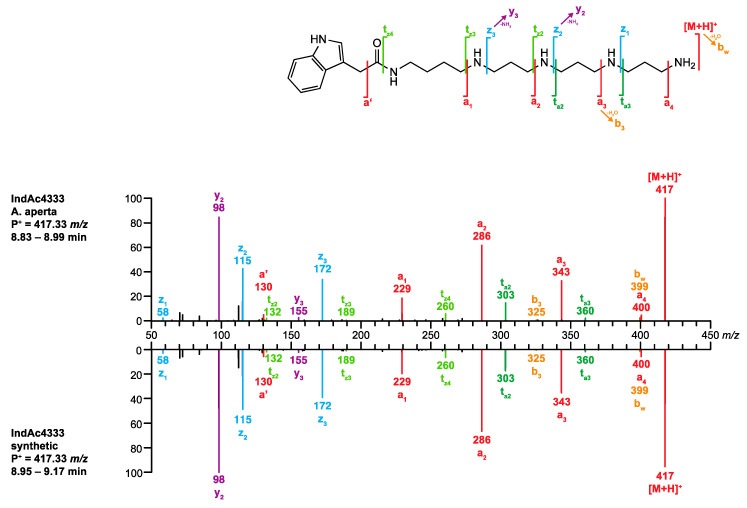
Comparison of the MS/MS spectrum of IndAc4333 obtained from *A. aperta* (**top**) venom with the spectrum of synthetic IndAc4333 (**bottom**).

**Figure 8 metabolites-10-00327-f008:**
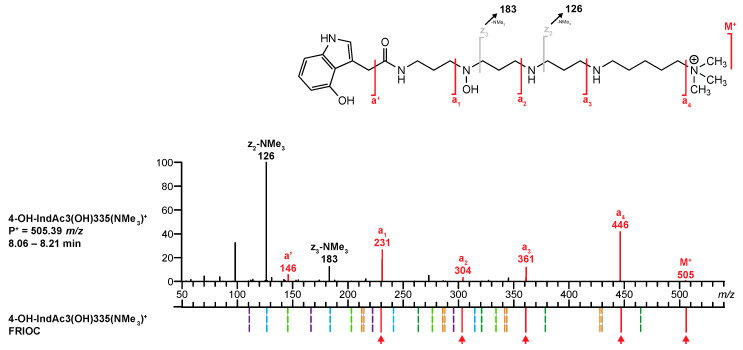
Comparison of the MS/MS spectrum of 4-OH-IndAc3(OH)335(NMe_3_)^+^ found in the venom of *A. aperta* with the fragments generated by the FRIOC tool. Spectra as well as measured and calculated values are shown in Appendix A.

**Figure 9 metabolites-10-00327-f009:**
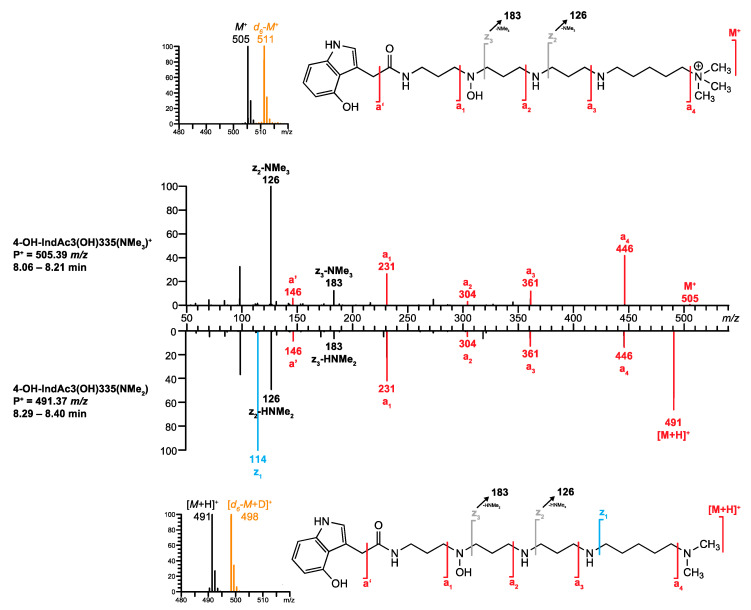
MS/MS spectra of 4-OH-IndAc3(OH)335(NMe_3_)^+^ and 4-OH-IndAc3(OH)335(NMe_2_) detected in *A. aperta* venom. Inserts corresponds to H/D exchange LC- MS with the mobile phases H_2_O/MeCN (black) and D_2_O/MeCN (orange), respectively. Spectra as well as measured and calculated values are shown in Appendix A.

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
