# Peer review of "VenoMS—A Website for the Low Molecular Mass Compounds in Spider Venoms"

_metabolites, 2020, doi:10.3390/metabo10080327_

Round 1

Reviewer 1 Report

This work describing the creation of a new computational tool for polyamines structure elucidation built off of venMS database shows great promise for advancing the field of toxinology and I highly recommend it for publications.

First and foremost, the lack of toxin databases information is extremely lacking in venom research, and in many species snakes, spiders, scorpions etc. Inasmuch as efforts in proteomics and transcriptomics have helped, the need for prediction tools has even greater. This manuscript addresses this need in a logical and sound approach for low MW polyamines.

I have only minor concerns for the manuscript:

  1. What is the function for these metabolites and why are they of interest currently?
  2. In the data curation, the authors use Scifinder and Google Scholar. NCBI and Uniprot have very large deposits of data on venom toxins, why not use these?
  3. the overall layout of this manuscript is odd due to the nature of content. In 2.2.1 Can S-5 and C-4 be illustrated in a graphic.
  4. On line 359, make this in line with Q6 and Q7 not inserted in Q10
  5. Figure 4 and5 could be combined, inserting the spectra and structure to show the utility of FRIOC. The examples will be more impactful and the manuscript more concise.
  6. Figure 6 should illustrate the use of FRIOC for Q10

Author Response

see pdf attached

Reviewer 2 Report

The manuscript submitted by Forster and colleagues described the implementation of a freely accessible MS database, named venoMS. This database is dedicated to low molecular weight metabolites present in spider venoms. Such a database can represent a useful resource for the scientists working in that field, it has also some additional interesting tools (such as FRIOC for calculating the expected fragment ion masses) and relevant metadata included. The manuscript is overall well written and the venoMS database is sufficiently innovative to extent the prior contributions. My concerns are mainly on the technical and methodological sides, which are - in my opinion – too poorly documented and explained to make the data fully convincing. Hence, I feel that major revision is necessary if the report is to advance this field.

Hopefully the following comments can be taken into account when submitting a revised version of the manuscript. Please do consider the following comments as friendly suggestions.

My concerns are mainly on the technical (mass spectrometry) side, which is - in my opinion – too poorly documented and discussed. This is typically the case for the procedures used for acquiring and annotating MS/MS spectra.

Although MS acquisition was performed using a high resolution mass spectrometer, ions are always labeled with nominal values (see Figures 2, 5, 7, 8, and 9). m/z ratios should be given with several digits after the decimal point, with corresponding mass accuracies reported.

It would be beneficial to add fragmentation conditions (resonant vs non-resonant, collision energy), since they will drastically influence the MS/MS pattern. When building spectral library, the use of “multiple collision energies” for data acquisition is highly recommended thus increasing the confidence in compound annotations by multiple independent verifications. It seems that the authors used a single collision energy (which is not given). Metabolite fragmentation is strongly collision energy dependent and thus no single collision energy can be applied to fit all metabolites. Standardizing or harmonizing fragmentation conditions is a prerequisite if the authors want to use relative intensities of fragment ions as identification criteria. Please elaborate on that aspect.

The authors should consider discussing and positioning their work with regard to the existing literature and standard practices from the metabolomics community on metabolite annotation and identification. Please refer to and discuss the papers of Sumner et al (Metabolomics 2007) and Schymanski et al (Environ Sci Technol 2014). Please also consider the recent papers of Oberacher et al (Metabolites 2019, Environ Sci Eur 2020) about tandem mass spectral libraries.

The annotation of ions and fragment ions in the mass spectra provided in both the document and venoMS website, should be improved and harmonized for all the metabolites present in the dabatase (e.g. acylpolyamines and small molecules). The use of a harmonized notation is also mandatory for interlaboratory MS data exchange and comparison (see Damont et al J Mass Spectrom 2019 as an illustration).

Specific comments

Lines 118-125: I think it would make sense that those reliability levels would be closer to the levels of metabolite identification proposed by the metabolomics society (Sumner et al, Metabolomics 2007). Harmonisation of identification levels/scales is mandatory in order to converge towards a single procedure that would be adopted by all, even if some minor adaptations may be specific to a given field of application.

Lines 161-162: In my opinion, the SAM2020 procedure should be detailed in the paper.

Lines 167-168: Is it possible to query the database directly using MS/MS data/spectra? Please elaborate on this aspect and provide additional detail on how the database is searchable.

Line 198: Relative intensities of fragment ions is highly dependent on the instrument used and fragmentation conditions applied. Proper comparison of fragment ion ratios might imply standardization of fragmentation conditions.

Lines 217-219: Are the retention times given relatively to a set of standards? Simply using the same chromatographic conditions in different laboratories might yield different retention times, if no prior system calibration is performed (e.g. due to different chromatographic dead volumes).

Lines 223-229: How is spectrum matching performed? Is it only based on visual inspection of spectra? It is now a common procedure in the metabolomics field to use dot product function or other similar methods for scoring the MS/MS spectral similarity and guarantee accurate identification. The authors should detail their methodology for evaluating the quality and similarity of MS/MS spectra.

Line 389: Please detail here the experimental procedures for acquiring data, so as to enable the reader to replicate them. Description in the website is not sufficient. Since the objective is to build a reference spectral library, the conditions for obtaining the spectra must be given very precisely. How were the MS/MS spectra acquired? Did the authors use DDA or targeted MS/MS workflows? What were the fragmentation conditions used? Etc…

Conclusion: Please also briefly discuss database sustainability. Would it be possible to download the database or contribute to its development by uploading new datasets?

Author Response

See attached pdf

Reviewer 3 Report

The article " venoMS – A Database for the Low Molecular Mass Compounds in Spider Venoms" is well presented for the journal. I would like the authors to get more  scientific publications included in the article. The conclusions and results could be expanded and could be more clear. The use of MS/MS to detect the spider venom is very interesting.

Author Response

See attached pdf

Round 2

Reviewer 2 Report

The authors have addressed some of the comments and concerns raised in the original review, but others are still pending.

Although the manuscript has been improved, I feel that some revisions are still required before publication.

I still believe that m/z ratios should be given in the Figures with several digits after the decimal point, with the corresponding mass accuracies. Mass accuracy is the first criterion for compound identification For instance, showing the nominal masses of the protonated and deuterated species in Figure 9 is not enough for me to ensure the identity of the ions and the exact number of deuterium atoms incorporated.

Lines 423-426: The authors now mention that deuterium-labeled TFA has been added to the mobile phases for H/D exchange. TFA can have some remnant effects, especially when using trapping devices such as the Orbitrap. This might lead to some H/D back exchanges and potentially misleading data interpretation. Please elaborate on this aspect.

Also, the authors argue in their response that they tried to respect the identification levels proposed by the Metabolomics Society and harmonize the procedure to annotate ion species. I really think this should be discussed in the revised paper by citing the references mentioned in my first review. Gathering both new and old data or heterogeneous data (MS and NMR) is also a point of interest that should be mentioned in the paper.

Last, line numbers cited in the rebuttal letter did not exactly match the changes made in the revised manuscript, so that checking expeditiously the changes made was not that easy.

Round 3

Reviewer 2 Report

The authors have addressed most of my comments. I think that the manuscript is now suitable for publication.

Please do consider that references 35 and 36 might have been inverted.

Author Response

Thank you for your comment. Citation numbers 35 and 36 have been corrected.